# Japan’s Practical Guidelines for Zinc Deficiency with a Particular Focus on Taste Disorders, Inflammatory Bowel Disease, and Liver Cirrhosis

**DOI:** 10.3390/ijms21082941

**Published:** 2020-04-22

**Authors:** Hiroko Kodama, Makoto Tanaka, Yuji Naito, Kazuhiro Katayama, Mitsuhiko Moriyama

**Affiliations:** 1Department of Health and Dietetics, Faculty of Health and Medical Sciences, Teikyo Heisei University, 2-51-4, Higashiikebukuro, Toshima-ku, Tokyo 170-8445, Japan; 2Department of Otorhinolaryngology-Head and Neck Surgery, Nihon University School of Medicine, Itabashi-ku, Tokyo 173-8610, Japan; tanaka.makoto39@nihon-u.ac.jp; 3Department of Gastroenterology and Hepatology, Graduate School of Medical Science, Kyoto Prefectural University of Medicine, Kyoto 620-8566, Japan; ynaito@koto.kpu-m.ac.jp; 4Department of Hepato-Biliary and Pancreatic Oncology, International Cancer Institute, Osaka 541-8567, Japan; katayama-ka@mc.pref.osaka.jp; 5Department of Internal Medicine, Nihon University School of Medicine, Itabashi-ku, Tokyo 173-8610 Japan; moriyama.mitsuhiko@nihon-u.ac.jp

**Keywords:** zinc deficiency, practical guideline, taste disorder, inflammatory bowel disease, chronic liver diseases

## Abstract

Zinc deficiency is common in Japan, yet awareness on this disorder is lacking. The Japanese Society of Clinical Nutrition recently issued the Japan’s Practical Guideline for Zinc Deficiency 2018 setting forth criteria for diagnosing zinc deficiency, i.e., (a) one or more symptoms of zinc deficiency or low serum alkaline phosphatase, (b) ruling out other diseases, (c) low serum zinc, and (d) alleviation of symptoms upon zinc administration. Serum zinc <60 μg/dL and 60–80 μg/dL indicate zinc deficiency and marginal deficiency, respectively. Zinc deficiency symptoms vary and include dermatitis and taste disorders among others. Zinc administration improves taste in 50–82% of patients suffering from taste disorders (a common symptom of zinc deficiency). Effects of zinc administration do not appear immediately, and therapy should be continued for at least three months. Zinc deficiency often accompanies various diseases and conditions. Here, we focus on inflammatory bowel diseases and liver cirrhosis. As zinc deficiency enhances intestinal inflammation via macrophage activation, we discuss the pathological mechanism for inflammation and zinc deficiency in the context of IBD. Zinc deficiency can also lead to a nitrogen metabolic disorder in patients with liver cirrhosis. Zinc supplementation can improve not only the ammonia metabolism, but also the protein metabolism. We also discuss directions for future studies of zinc deficiency.

## 1. Introduction

Zinc deficiency is common in developing countries and affects over two billion people worldwide [1]. Zinc deficiency is common in Japan as well. Wessells and Brown reported that the estimated percentage of the population with inadequate zinc intake in Japan in 1990, 1995, and 2000 was 14.7, 14.7, and 15.5%, respectively, indicating an increasing prevalence in zinc deficiency [2]. One reason for this is the rapidly growing population of the elderly, who suffer from chronic diseases and thus are treated with medications. Another contributor is the lack of awareness among Japanese clinicians about zinc deficiency. Against this backdrop, the Japanese Society of Clinical Nutrition issued the Practical Guideline for Zinc Deficiency (hereinafter referred to as the “Practical Guideline”) in 2018 [3].

In this review article, we start by providing a condensed summary of the Practical Guideline. Symptoms and signs of zinc deficiency vary and include taste disorders, dermatitis, hair loss, loss of appetite, susceptibility to infection, and hypogonadism. Taste disorder, a typical symptom of zinc deficiency, is discussed in further detail in Section 3. Chronic diseases, including chronic hepatic diseases and inflammatory bowel diseases, often accompany zinc deficiency and will be discussed in Section 4 and Section 5. Symptoms of many of these diseases are likely to improve with zinc therapy.

## 2. Japan’s Practical Guideline for Zinc Deficiency

### 2.1. Prevalence of Insufficient Zinc Intake and Zinc Deficiency in Japan

As the prevalence of insufficient zinc intake seems to differ by age and gender, we examined the percentage of insufficient zinc intake by age bracket and gender. Figure 1 shows the percentage of Japanese people with insufficient zinc intake, i.e., below the recommended dietary allowance (RDA) [4,5]. Zinc intake is insufficient in 60–70% of both males and females aged ≥20 years and is significantly lower than the RDA in pregnant and lactating women as well (Figure 2) [4,5]. Zinc deficiency is known often to accompany severe physical disabilities, liver cirrhosis, chronic hepatitis, chronic inflammatory bowel diseases, type 2 diabetes mellitus, chronic kidney disease, cardiac insufficiency, short stature, and is also more common in the elderly and professional athletes [6,7,8,9,10,11,12,13,14,15].

### 2.2. Causes of Zinc Deficiency in Japan

#### Causes of Zinc Deficiency Can Be Divided into Four Major Categories

One cause is insufficient zinc intake. Meats are rich in zinc, while vegetables only have minimal amounts of zinc (Food Data Chart–Zinc (http://apjcn.nhri.org.tw/server/info/booksphds/books/foodfacts/html/data/data5j.html)). Since the elderly generally eat less food, particularly meat [5], they often succumb to many nutritional deficiencies, including zinc deficiency. Infants may also suffer from zinc deficiency if the mother’s breastmilk has a low zinc content. Although rare, some mothers have mutations in the *ZnT2* gene, which prevents the transfer of zinc from mammary gland cells to breast milk [16]. Severely handicapped patients who eat little food may also suffer from zinc deficiency.

Another cause is interference with zinc absorption. This can result from chronic liver diseases, cirrhosis, chronic inflammatory bowel diseases, excess intake of phytic acid, and other causes.

Yet another cause is an increase in the required amount of zinc. The RDA of zinc in pregnant women (which applies to lactating women as well) is higher than that in non-pregnant women. Yet, as mentioned above, the RDA and the mean zinc intake in pregnant and lactating women reveal insufficient zinc intake in these women [17]. Athletes, especially women, are also likely to experience zinc deficiency [18].

Finally, excessive excretion of zinc via urine can cause zinc deficiency. Excessive excretion occurs in those with conditions such as chronic kidney disease [12,13] and diabetes mellitus [11].

### 2.3. Diagnostic Guideline for Zinc Deficiency

Table 1 summarizes the four criteria in the Practical Guideline for diagnosing zinc deficiency. The first criterion is the existence of one or more symptoms of zinc deficiency or a low level of serum alkaline phosphatase (ALP), a zinc-dependent enzyme. While low serum ALP is listed as a criterion, ALP levels can be inherently high in patients with liver disease, osteoporosis, chronic kidney disease, and diabetes mellitus. Thus, serum ALP levels are not used as a diagnostic criterion for patients with these specific diseases.

The second criterion is the ruling out of other diseases which are also associated with the abovementioned symptoms. The third criterion is a low serum zinc level, and the fourth criterion is the ability to improve symptoms with zinc administration. Zinc treatment is recommended as acceptable under the first, second, and third criterion.

The most important criterion for diagnosing zinc deficiency is serum zinc levels. In Harrison’s Principles of Internal Medicine, a serum zinc level <70 µg/dL is listed as a criterion for zinc deficiency [19]. However, the zinc clearance test is reportedly more accurate for diagnosing the disorder than serum zinc levels, especially in cases of marginal zinc deficiency [20,21,22,23]. Nakamura et al. [22] reported that 10 patients with short stature who were diagnosed with zinc deficiency using this test showed improved stature upon zinc treatment. In these patients, the mean serum zinc level prior to zinc administration was 78 ± 6 µg/dL (mean ± SD), suggesting that those with serum zinc ≥70 µg/dL are also susceptible to zinc deficiency. In another study, Kaji et al. reported a mean serum zinc level of 75.0 ± 12.7 µg/dL in patients with short stature who were diagnosed with zinc deficiency with the zinc clearance test [23]. Tomita et al. proposed 80 µg/dL as a cutoff level for diagnosing zinc deficiency [24].

Based on the above studies, the Practical Guideline proposed that a serum zinc level < 60μg/dL indicates zinc deficiency, 60–80 μg/dL indicates marginal zinc deficiency, and >80 μg/dL indicates the normal zinc status. The Practical Guideline recommends that zinc deficiency is diagnosed based on the first three criteria and treated accordingly. However, as mentioned above, serum zinc levels may not accurately reflect zinc deficiency. Thus, more sensitive and accurate biomarkers should be identified to improve the accuracy of zinc deficiency diagnoses.

To our knowledge, only one other report has been published on practical guidelines for zinc deficiency; namely, the BMJ Best Practice of Zinc Deficiency [25], which reports in detail on zinc deficiency, including symptoms, for serum zinc levels <60 µg/dL or 70 µg/dL. In this report, we also note that, due to the relatively low sensitivity of serum zinc levels in cases of marginal deficiency, oral supplementation should be considered if symptoms are typical, even if test results are normal; this is consistent with our diagnostic guidelines.

### 2.4. Treatment of Zinc Deficiency

Patients with zinc deficiency are initially recommended to consume zinc-enriched foods, e.g., those listed in Table 2. Oysters contain a large amount of zinc, as do meats and scallops. Liver is particularly rich in zinc. Japanese foods, including tofu, rice, and fermented soybeans (natto), also contain large amounts of zinc.

Although recommended above, zinc deficiency is difficult to overcome by dietary therapy alone. When dietary therapy is insufficient, the Practical Guideline recommends administering zinc to patients with zinc deficiency at a dosage of 50–100 mg/day for adults and 1–3 mg/kg/day for children.

Serum zinc levels are often low in patients with some chronic diseases. When such patients present with low serum zinc levels, their condition often improves with zinc therapy [26,27,28,29,30,31]. Thus, zinc therapy is recommended for patients with low serum zinc levels, even if they do not present with symptoms of zinc deficiency.

The Practical Guideline also describes potential adverse effects of zinc therapy, including nausea, vomiting, and itching. In some cases, nausea and vomiting can be prevented by taking zinc after meals. Another potential adverse effect is copper deficiency, which is associated with anemia and leukopenia [32,33]. Serum levels of both zinc and copper should be examined once every three or four months during zinc therapy.

Zinc is administered to patients suffering from a wide variety of diseases, including chronic liver disease, cirrhosis, IBD, chronic kidney disease [12,13], and diabetes mellitus [11]. It is also administered to those taking medications with chelating effects, and to athletes and pregnant women. Further study of effects of zinc administration in each patient population is warranted.

## 3. Zinc Deficiency and Taste Disorders

Taste sensation allows one to sense important information related to survival. In other words, it allows one to judge whether something is edible or not. This sensation is detected by receptors that categorize taste into the following: sweet, salty, sour, bitter, and umami.

### 3.1. Taste Perception System

A water-soluble chemical substance that presents a taste in food is received by taste receptors on taste cells of taste buds in the mucous membrane of the tongue and pharynx, and is projected to the gustatory area of the cerebral cortex via taste nerves. There are roughly 7000 taste buds (peripheral receptors of taste) in the oral cavity, pharynx, and larynx, with a particularly high concentration in the lingual papillae on the tongue surface. The five basic tastes are sensed by different taste receptors on taste cells.

The type of taste nerve varies by location inside the oral cavity. Taste information is transmitted from the left and right chorda tympani nerve, large pyramidal nerve, glossopharyngeal nerve, and vagus nerve to the cerebral cortex’s gustatory area via the medullary solitary nucleus and thalamus.

### 3.2. Effects of Zinc Deficiency on Taste Function in Rats

Rats fed a zinc-deficient diet begin to drink bitter water, which would typically be avoided, in addition to normal water. This suggests that they suffer from a taste disorder, a common symptom of zinc deficiency [34].

In taste cells of rats fed a zinc-deficient diet, microstructural abnormalities such as microvilli rupture and vacuolation are observed by electron microscopy [35]. Cell differentiation from basal cells is also impaired, and the time required for cell turnover is prolonged [36]. Studies in rats have also found that reduced expression of TAS2R40 and TAS2R107, bitter taste receptors, is evident in zinc deficiency [37].

### 3.3. Taste Disorders

Taste disorders reflect an abnormality in taste perception. According to a patient survey conducted by the Japanese Society of Stomato-pharyngology in 2003, roughly 240,000 patients suffer from taste disorders, representing a 1.7-fold increase since 1990. Taste disorders affect many elderly and are more common in women, with a male-to-female ratio of 2:3. The number of affected patients is expected to rise as the population ages.

Symptoms of taste disorders can be divided into two categories: quantitative dysgeusia and qualitative dysgeusia (Table 3). Many types of taste disorders are known, including zinc deficiency taste disorder, idiopathic taste disorder, drug-induced taste disorder, psychogenic taste disorder, and systemic taste disorder. The zinc deficiency taste disorder has no clear cause or trigger other than a reduction in serum zinc levels. The idiopathic taste disorder with no apparent cause is likely associated with zinc deficiency.

### 3.4. Zinc Replacement Therapy

Henkin et al. first reported the treatment of taste disorders by oral administration of zinc in 1970 [38]. In that report, drug-induced taste disorder caused by d-penicillamine was successfully treated by oral administration of zinc chloride.

In 1991, Yoshida reported an 82% improvement in the zinc deficiency taste disorder and the idiopathic taste disorder in a double-blind comparative test using 67.5 mg zinc gluconate [39]. In other reports, Sakai reported a 76% improvement using 87 mg zinc picolinate [40], and Heckmann reported a 50% improvement using 20 mg zinc gluconate [41]. Zinc gluconate and zinc picolinate are not commonly used in Japan, because they require drug preparation at each individual clinic. Therefore, a daily dose of Polaprezinc used to treat gastric ulcers and containing 34 mg of zinc is prescribed to patients with taste disorders in Japan. Sakagami reported an 80% improvement in subjective taste disorder symptoms using 150 mg Polaprezinc (Table 4) [42].

Notably, zinc replacement therapy is not immediately effective. In a double-blind comparative study using Polaprezinc, the efficacy was found to be 13.6% in the 4th week of treatment, but this had increased to 47.7% in the 12th week and 58.8% in the 24th week of treatment. Thus, the longer the treatment period, the stronger the efficacy. Zinc replacement therapy should be continued for at least three months [42].

Zinc acetate hydrate used to treat Wilson’s disease has been eligible for health insurance coverage for the treatment of hypozincemia in Japan since 2017. This drug is expected to be effective for treating taste disorders as well.

### 3.5. Nutritional Guidance

Nutritional guidance is also effective for treating taste disorders. The standard amount of zinc intake recommended by the Ministry of Health, Labor and Welfare in Japan is 10 mg per day for men and 8 mg per day for women. The actual intake amount, however, is below the recommended standard, at 8.9 mg per day for men and 7.3 mg per day for women. Indeed, half of the Japanese population is at risk of zinc deficiency taste disorder. Therefore, at our facility, we show patients a list of foods that are either rich in zinc, promote zinc absorption, or inhibit zinc absorption, and encourage them to be more aware of their daily eating habits.

## 4. Zinc Deficiency Enhances Intestinal Inflammation via Macrophage Activation

Zinc deficiency is a very common symptomatic condition. According to a multicenter hospital-based case–control study conducted in Japan, higher doses of zinc taken by patients one year before the study period were associated with a decreased odds ratio for ulcerative colitis (UC), indicating that high zinc intake can protect against UC [43]. In a prospectively collected inflammatory bowel disease (IBD) registry study, zinc deficiency was associated with an increased risk of subsequent hospitalization, surgery, and disease-related complications in patients with IBD, Crohn’s disease (CD), and UC [8]. We measured serum zinc levels and examined the correlation with clinical activity in patients with CD. As shown in Figure 3, serum zinc levels were below 80 µg/dL in more than 75% of CD patients, and the Pearson’s correlation analysis revealed that serum zinc concentrations were negatively (*r* = −0.75, *p* < 0.05) correlated with the Harvey–Bradshaw Index (HBI), an index of clinical activity, in patients with CD [44]. Despite the abovementioned results, the detailed mechanism by which zinc deficiency enhances the inflammatory response remains unclear. In this section, we have demonstrated that zinc deficiency aggravates colonic inflammation through the activation of T helper 17 (Th17) cells, and that production of interleukin-23p19 (IL-23p19) using zinc-deficient macrophages is crucial for Th17 cell activation.

### 4.1. Role of Metallothioneins (MTs) in Intestinal Inflammation

MTs are a family of low molecular weight proteins containing cysteine residues, which enable high-affinity binding to monovalent and divalent heavy metal atoms. We previously reported that production of several inflammatory cytokines (TNF-α, IFN-γ, and IL-17) was significantly elevated in peritoneal macrophages derived from MT-I/II knockout mice [45], and that dextran sulfate sodium (DSS)-induced colonic inflammation as an animal model of UC was aggravated significantly in knockout mice [45]. Our results also confirmed a significantly higher expression of MT-I/II in the colonic mucosa of the DSS-induced colitis model using the real-time PCR analysis. In addition, MT-positive cells were detected in the lamina propria and the submucosal layer by immunohistochemical and immunofluorescence staining, and were mainly co-localized with F4/80-positive macrophages [44]. In contrast, MT-positive cells were not detected in the colon of MT-I/II knockout mice before and after DSS administration. These results suggest that MTs protect against colonic mucosal inflammation in the mouse model of DSS colitis via their anti-inflammatory function in macrophages, indicating that endogenous MTs, likely by binding to intracellular zinc, play an important role in protecting the intestinal mucosa.

### 4.2. Zinc Deficiency Aggravates Trinitrobenzene Sulfonic Acid (TNBS)-Induced Colitis

To assess effects of zinc deficiency on colonic inflammation in mice treated with 2,4,6-trinitrobenzene sulfonic acid (TNBS), the mice were intraperitoneally administered with N,N,N′,N′-tetrakis(2-pyridylmethyl)ethylenediamine (TPEN), a zinc-chelating reagent, 24 h before TNBS administration. TPEN administration led to a significant reduction in serum zinc levels compared to the mice administered a vehicle. TNBS-induced colitis was significantly aggravated in the TPEN-treated mice, and the production of IL-17A, but not IFN-γ or IL-10, was enhanced significantly in the TPEN + TNBS-treated mice [44]. Flow cytometry further revealed that the proportion of IL-17A-producing CD4^+^ cells (Th17) was increased significantly by TPEN administration in lamina propria mononuclear cells (LPMCs) obtained from the inflamed colon tissue, but not in cells from the spleen or the mesenteric lymph node [44]. However, an in vitro differentiation study showed that TPEN-mediated zinc deficiency did not affect Th17 differentiation from naïve CD4^+^ cells, suggesting that Th17 activation mediated by zinc deficiency requires involvement of immune cells other than CD4^+^ cells. These results suggest that zinc deficiency-mediated aggravation of colitis is caused by changes in the function of immune cells, particularly of T cells.

### 4.3. Effects of Zinc Deficiency on Macrophage Phenotype under Inflammatory Conditions

Inflammatory macrophages contribute to the establishment of an environment for a Th17 response by secreting several cytokines. This suggests that the skewing of the macrophage differentiation may be a mechanism by which TPEN-induced zinc deficiency induces a Th17 shift. To test this hypothesis, we isolated LPMCs from inflamed colons and then evaluated the proportion of M1- and M2-type macrophages using flow cytometry. Figure 4 shows that TPEN administration significantly increased the proportion of Ly6C^+^ cells (M1 macrophages) in intestinal CD45^+^CD11b^+^ macrophages [44]. Moreover, TPEN administration significantly decreased the proportion of CD206^+^ cells (M2 macrophages) and markedly increased the M1/M2 ratio. In addition, we confirmed that the expression of IL-23p19, but not of other cytokines, was significantly enhanced in the TPEN + TNBS-treated mice compared to the TNBS-treated mice, and TPEN administration significantly increased the proportion of IL-23p19^+^ cells in intestinal macrophages [44].

Next, we investigated the relationship between macrophage polarization and the abundance of intracellular zinc. TPEN-mediated zinc depletion in bone marrow-derived macrophages (BMDMs) significantly upregulated IL-23p19 expression that was concentration- and time-dependent. This induction was reversed by zinc sulfate treatment [44]. Flow cytometry analyses using BMDMs showed that the combination of IFN-γ + lipopolysaccharide (LPS) caused M1 skewing through a reduction in intracellular zinc. We also found IL-17A production to be significantly greater when CD4^+^ cells were cultured in the conditioned medium derived from TPEN-treated BMDMs, and that the addition of a neutralizing antibody against IL-23p19 suppressed IL-17A production by CD4^+^ cells cultured in the conditioned medium. Finally, the nuclear distribution of interferon regulatory factor 5 (IRF5), an important transcriptional factor for IL-23p19, induced by zinc deficiency enhanced IL-23p19 expression in TPEN-treated BMDMs. These data indicated that intracellular free zinc plays a crucial role in the induction of IL-23p19 from activated macrophages [44].

In clinical practice, Schmidt et al. [46] reported that IL-23p19 expression was strongly upregulated in colonic mucosa in patients with CD, but not in patients with UC. In fact, blocking the IL-23 pathway using ustekinumab is an effective therapeutic strategy for human CD patients [47]. We showed that intracellular zinc status is closely associated with IL-23p19 expression in macrophages. If zinc status also affects IL-23p19 regulation in CD patients, then close monitoring of serum zinc levels might be useful for the management of such patients, especially those treated with ustekinumab.

Inconsistent with our findings, Kido et al. [48] demonstrated that zinc deficiency-induced aggravation of inflammation is related to Th2 lymphocytes and followed by the loss of GATA-3 and IL-4 expression, as well as of anti-inflammatory M2 macrophages (Figure 5). The decrease in the M2 macrophage population due to zinc deficiency was also confirmed in the TNBS-induced colitis by our studies [44].

In conclusion, zinc deficiency aggravates inflammation by activating IRF/IL-23-mediated M1 macrophages and inhibiting GATA-3/IL-4 mediated M2 macrophages (Figure 4). Although these data pertain only to the murine experimental colitis, we believe that they provide foundational support for a potential randomized controlled trial aimed to evaluate therapeutic efficacy of zinc supplementation in IBD. Our results collectively suggest that intracellular zinc acts as a signaling molecule and is involved in IL-23 production in activated macrophages. Further studies examining efficacy of anti-IL-23 antibodies against IBD would be informative.

## 5. Zinc and Protein Metabolism in Chronic Liver Disease

Liver cirrhosis is a late-stage chronic liver disease that can cause various complications due to decreased hepatic function and portal hypertension. The liver is the principal organ responsible for metabolizing nutrients. Thus, liver cirrhosis leads to defective nutrient metabolism, which in turn can cause complications and worsen prognosis.

Of the three major categories of nutrient metabolism (protein, lipid, glucose), protein metabolism plays an important role in albumin and prothrombin synthesis and in ammonia detoxification. These processes are related to hepatic functional reserve, play a role in the development of hepatic encephalopathy, and are targets of nutrition therapy for liver cirrhosis.

Zinc plays important roles in the activation and structural maintenance of as many as 300 proteins and enzymes that contribute to various biological processes, including growth, antioxidant effects, immune response, apoptosis, aging, and carcinogenesis [49]. Ornithine transcarbamylase, a zinc enzyme involved in the urea cycle, plays an important role in ammonia metabolism [50]. Several studies have shown that zinc deficiency is common in patients with cirrhosis and that nitrogen metabolism is affected by zinc deficiency [51,52]. We have recently reviewed zinc and protein metabolism in chronic liver disease [53], and have discussed these findings briefly in the present review.

### 5.1. Mechanism Underlying Reduced Capacity to Detoxify Ammonia in Patients with Liver Cirrhosis

Ornithine transcarbamylase, which plays an essential role in the urea cycle, is a zinc-dependent enzyme, the activity of which decreases under conditions of zinc deficiency. Zinc deficiency is prevalent in patients with liver cirrhosis and could be a reason underlying the reduced capacity of the urea cycle in these patients.

We recently reported on the prevalence and implications of zinc deficiency in patients with chronic liver disease. In 2009, 299 patients with liver cirrhosis who did not have a hepatocellular carcinoma were enrolled from 14 medical institutes in Japan. Of these 299 patients, 235 had not received any oral zinc preparations.

Figure 6 shows the relationship between serum zinc levels and the Child–Pugh classification. Serum zinc levels in patients with Child–Pugh class B and those with Child–Pugh class C cirrhosis were significantly lower than those in patients with Child–Pugh class A cirrhosis [52].

Figure 7 shows the correlation between levels of serum zinc and those of albumin and ammonia, as well as the branched-chain amino acid to tyrosine ratio (BTR). Serum albumin levels showed a significant positive correlation with serum zinc levels, as did the BTR, while serum ammonia levels showed a significant negative correlation with serum zinc levels.

Zinc deficiency in patients with liver cirrhosis decreases the ability of the liver to detoxify ammonia, resulting in a compensatory increase in ammonia detoxification by skeletal muscles. This in turn leads to increased consumption of branched-chain amino acids (BCAAs) by skeletal muscle, resulting in BCAA deficiency in serum and a decreased ability of the liver to synthesize albumin. This cascade leads to the worsening of the liver function.

### 5.2. Does Zinc Supplementation in Patients with Liver Cirrhosis Improve Protein Metabolism and Prognosis?

Several studies have demonstrated the beneficial effects of zinc supplementation on hyperammonemia associated with liver cirrhosis. Figure 8 shows data from one such study, in which 12 patients with liver cirrhosis, hyperammonemia, and hypozincemia were randomly assigned to receive either zinc supplementation or a placebo. Zinc acetate (150 mg/day) was administered for three months. Figure 8 shows changes in serum zinc levels and the percentage change in serum ammonia levels. Serum zinc levels increased significantly, while serum ammonia levels decreased significantly in the patients receiving zinc supplementation compared with the control group.

Other studies have also demonstrated benefits of zinc supplementation. For instance, Reding et al. reported that zinc supplementation was effective for treating hepatic encephalopathy in patients with liver cirrhosis [55]. Marchesini et al. demonstrated that zinc supplementation in patients with liver cirrhosis resulted in improved encephalopathy indicators, such as better scores on the number connection test, significantly decreased blood ammonia levels, and slightly improved Fisher ratios [56]. They also found that the metabolism of alanine to urea nitrogen improved with zinc supplementation.

## 6. Conclusions and Future Directions

The prevalence of zinc deficiency in Japan is the highest among the developed countries, likely due to the increasing population of the elderly who suffer from and take medications for chronic diseases. Zinc deficiency can accompany various conditions and diseases, and can affect the elderly, pregnant women, lactating women, patients with chronic diseases, such as diabetes mellitus, liver disease, kidney disease, inflammatory bowel disease, and during the administration of chelators. Nonetheless, zinc deficiency is often missed in clinical settings in Japan given the relative lack of awareness of this disorder.

The Japanese Practical Guideline for zinc deficiency was issued. Although serum zinc level is one criterion, serum zinc levels do not appear to accurately reflect zinc deficiency. Accordingly, more sensitive and accurate biomarkers of zinc deficiency are needed. The number of patients with taste disorders, a typical symptom of zinc deficiency, has also increased in Japan. Zinc administration alleviates not only zinc deficiency-induced taste disorders, but also idiopathic, drug-induced, and systemic taste disorders. The pathophysiological mechanisms should be examined further, with a specific focus on the effect of zinc administration on these taste disorders.

Zinc deficiency is often observed in patients with inflammatory bowel diseases and liver cirrhosis, as these conditions interfere with zinc absorption. Here, we found that zinc deficiency enhances intestinal inflammation via macrophage activation in patients with inflammatory bowel diseases. However, the detailed mechanisms by which zinc deficiency enhances the inflammatory response remains unclear. The activity of ornithine transcarbamylase, a zinc-dependent enzyme, decreases under conditions of zinc deficiency such that the body’s capacity to detoxify ammonia in the liver is lowered, worsening the condition of the liver. 

The Practical Guideline recommends zinc therapy for treating zinc deficiency. Supporting this recommendation are numerous studies demonstrating the beneficial effects of zinc supplementation in patients with various diseases and conditions. Notwithstanding, the effects of zinc administration will need to be verified for each individual disease and condition. 

## Figures and Tables

**Figure 1 ijms-21-02941-f001:**
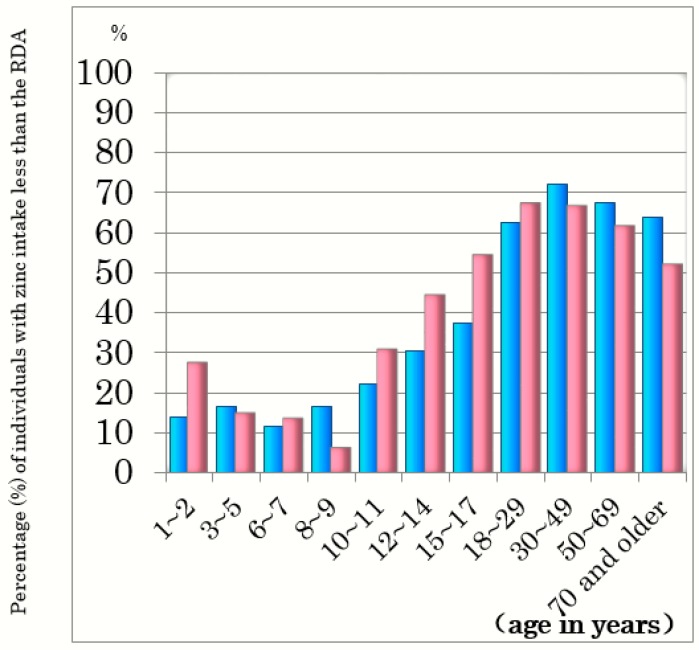
Percentage of people with zinc intake less than the RDA in Japan. The recommended dietary allowance (RDA) of zinc was obtained from the Dietary Reference Intakes for Japanese 2020 [4]. The data for zinc intake were obtained from the National Health and Nutrition Survey 2017 in Japan [5]. The percentage of individuals zinc intake less than the RDA was calculated from the data of RDA (Ref. [4]) and the zinc intake (Ref. [5]). Blue and red bars are those for males and females, respectively.

**Figure 2 ijms-21-02941-f002:**
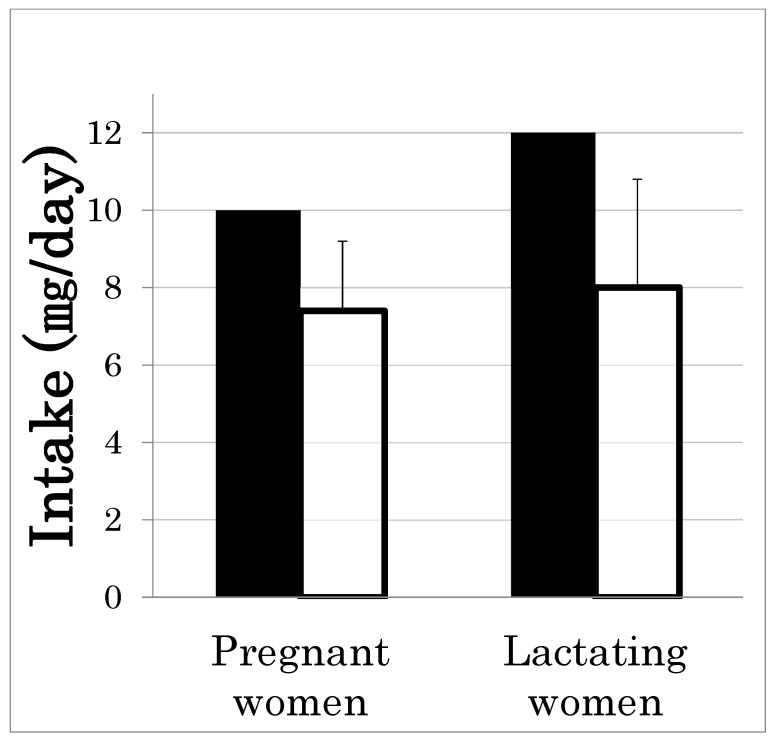
The comparison between the RDA and the actual zinc intake (mean ± SD) in pregnant or lactating women in Japan. The recommended dietary allowance (RDA) of zinc was obtained from the Dietary Reference Intakes for Japanese 2020 [4]. The RDA was 10 and 12 mg/day for pregnant and lactating women, respectively. The data for zinc intake were obtained from the National Health and Nutrition Survey 2017 in Japan [5]. The mean ± the SD of intake of pregnant and lactating women were 7.4 ± 1.8 mg/day and 8.0 ± 2.8 mg/day, respectively. Black bars indicate the RDA and white bars indicate the mean zinc intake.

**Figure 3 ijms-21-02941-f003:**
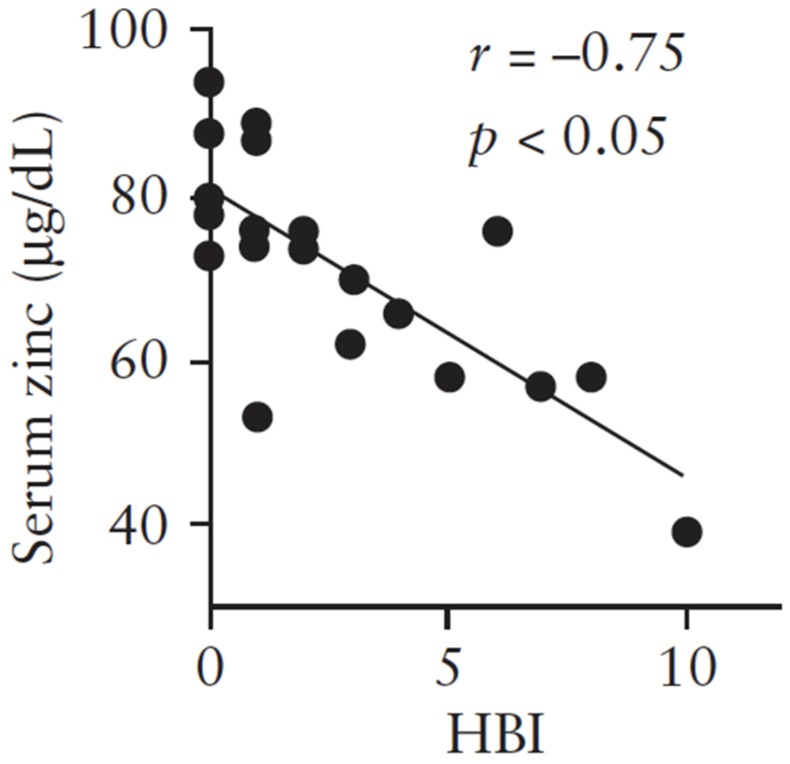
The association between serum zinc levels and clinical activity in patients with Crohn’s disease. HBI, Harvey–Bradshaw Index, an index of clinical activity for Crohn’s disease (*n* = 22). Cited from Higashimura et al. [44]. (used with permission).

**Figure 4 ijms-21-02941-f004:**
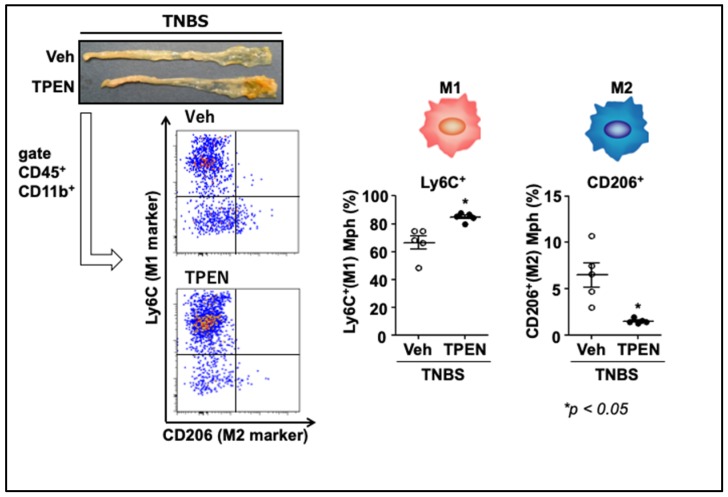
Effects of zinc deficiency on the macrophage phenotype under inflammatory conditions. Proportions of Ly6C^+^ macrophages (M1) and CD206^+^ macrophages (M2) in colonic lamina propria mononuclear cells (LPMCs) as determined by flow cytometry. Veh; vehicle treated group, TPEN; N,N,N′,N′-tetrakis(2-pyridylmethyl)ethylenediamine (TPEN)-treated group. The data are presented as the mean ± the standard error of the mean of five mice. Cited from Higashimura et al. [44]. (used with permission)

**Figure 5 ijms-21-02941-f005:**
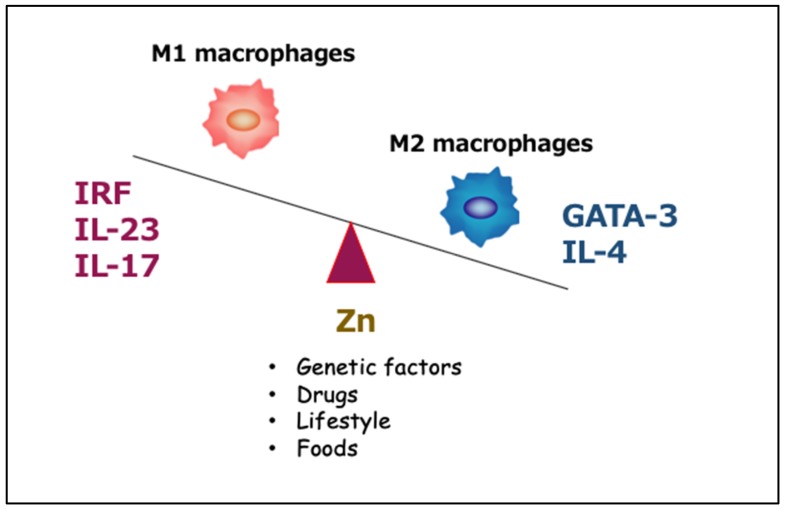
Zinc-mediated regulation of M1/M2 macrophage activation. In zinc-deficient macrophages, interferon-regulatory factor 5 (IRF5) translocates into the nucleus and induces IL-23p19 expression, thereby promoting Th17 differentiation from naïve T cells. Zinc deficiency-induced aggravation of inflammation is associated with a loss of GATA-3 and IL-4 expression and anti-inflammatory M2 macrophages.

**Figure 6 ijms-21-02941-f006:**
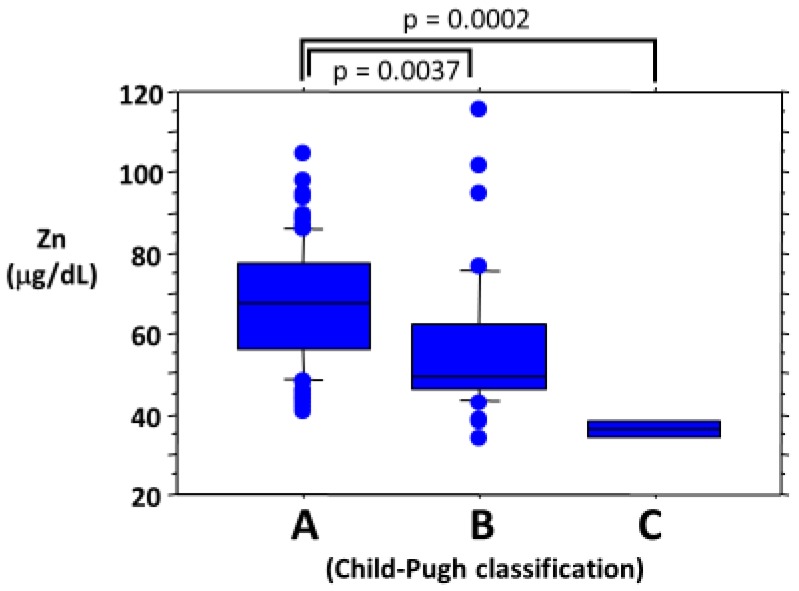
Relationship between serum zinc levels and Child–Pugh classification. A, Child-Pugh class A; B, Child-Pugh class B.; C, Child-Pugh class C.

**Figure 7 ijms-21-02941-f007:**
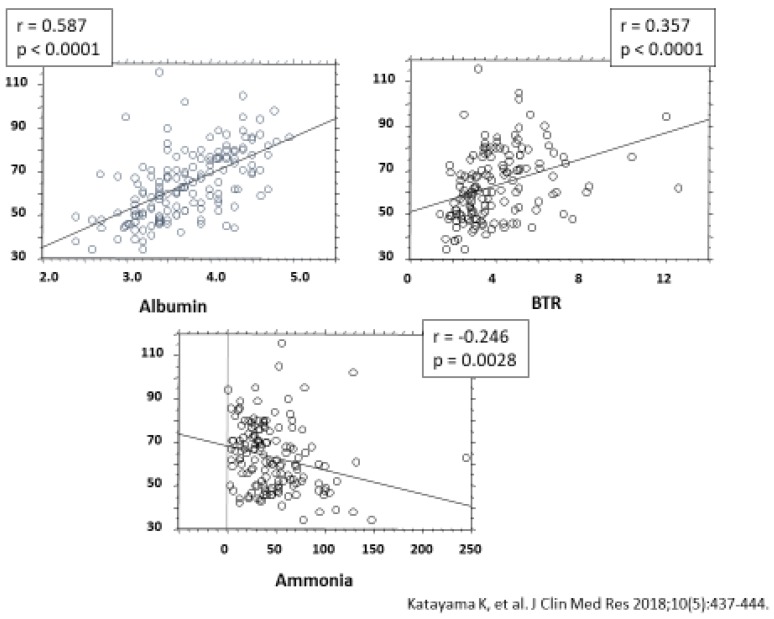
Useful effects of zinc supplementation on hyperammonemia associated with liver cirrhosis. Cited from Katayama et al. [52].

**Figure 8 ijms-21-02941-f008:**
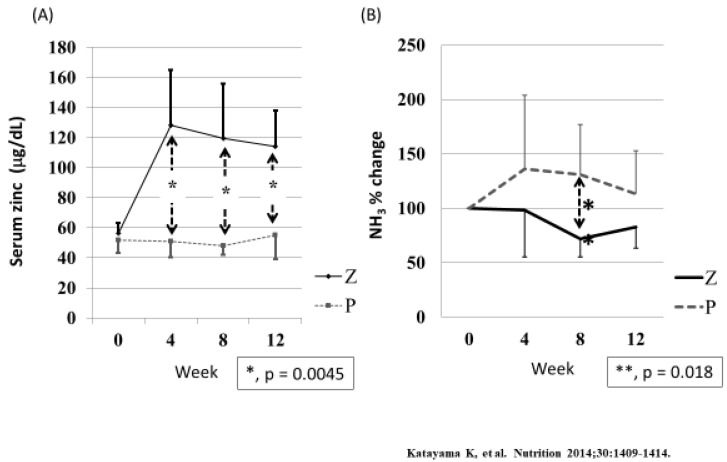
The effect of zinc supplementation on serum ammonia levels in patients with liver cirrhosis. Cited from Katayama et al. [54]. Twelve patients with liver cirrhosis, hyperammonemia, and hypozincemia were randomly assigned to receive either zinc supplementation or a placebo. Z = zinc supplementation; P = placebo. (**A**) Serum zinc levels of patients in the Z group rose significantly (*p* = 0.0037, Friedman test) during the trial, whereas those in the P group did not change significantly (*p* = 0.4402, Friedman test). Intergroup differences were significant by Mann-Whitney test at week 4, 8 and 12.; (**B**) Relative serum ammonia levels of patients in the Z group fell significantly (*p* = 0.0114, Friedman test), whereas those in the P group did not (*p* = 0.4717, Friedman test). The mean value of Z group was significantly lower than that of P group (*p* = 0.0188, Mann-Whitney test) at week 8.

**Table 1 ijms-21-02941-t001:** The diagnostic guideline for zinc deficiency.

**Zinc deficiency can be reliably diagnosed by the four criteria below:**
I. One or more symptoms and signs of zinc deficiency (dermatitis, aphthous stomatitis, hair loss, loss of appetite, taste disorder, hypogonadism in males, anemia, increased susceptibility to infection, growth disturbances in terms of weight and height in children, and low levels of serum alkaline phosphatase (ALP). However, serum ALP levels are not always low in patients with liver disease, osteoporosis, chronic kidney disease, or diabetes mellitus.
II. Ruling out of other diseases associated with the above symptoms or low serum ALP levels. For example, conditions such as contact dermatitis, atopic dermatitis, dermatitis due to deficiencies in vitamin A, biotin, or essential fatty acids, alopecia areata, hair-pulling, short stature due to growth hormone deficiency, familial short stature, Turner syndrome, and congenital hypophosphatasia should be ruled out.
III. Low serum zinc levels
III-1: <60 µg/dL: zinc deficiency
III-2: 60–80 µg/dL: marginal zinc deficiency
(Blood sampling is recommended in the morning under fasting conditions)
IV. Zinc treatment can be performed on patients who meet criteria I, II, and III. Symptoms in these patients can be improved with zinc treatment.

**Table 2 ijms-21-02941-t002:** Foods rich in zinc.

Food	Zinc Concentration (mg/100 g)	Amount of Zinc/Dish	
		Weight	Zinc Concentration (mg)
Oyster	13.2	5 pieces (60 g)	7.9
Pig liver	6.9	70 g	4.8
Flat iron beef steak	5.8	70 g	4.1
Beef liver	3.8	70 g	2.7
Chicken liver	3.3	1 meal (70 g)	2.3
Beef flank	3.0	1 meal (70 g)	2.1
Scallops	2.7	3 pieces (60 g)	1.6
Rice	0.8	150 g	1.2
Grilled eel	1.4	Half of an eel (80 g)	1.1
Firm tofu	0.6	Half (150 g)	0.9
Natto	1.9	1 piece (40 g)	0.8
Egg yolk	4.2	1 piece (16 g)	0.7
Japanese noodles	0.4	1 dish (180 g)	0.7
Processed cheese	3.2	1 piece (20 g)	0.6
Powdered green tea	6.3	1 spoonful (6 g)	0.4
Cocoa	7.0	1 spoonful (6 g)	0.4

Calculated using the data in Standard Tables of Food Composition I (Japan, 2015) as published by the Ministry of Education, Culture, Sports, Science and Technology in Japan.

**Table 3 ijms-21-02941-t003:** Symptoms of taste disorders.

Quantitative Dysgeusia
taste reduction
taste loss
loss of a particular taste
**Qualitative Dysgeusia**
spontaneous dysgeusia
feeling that food tastes different
bad taste regardless of what is eaten

**Table 4 ijms-21-02941-t004:** Effects of zinc administration on zinc deficiency and idiopathic taste disorders.

	Mean Age	*n*	Zinc Prepararion Dosage (mg/Day)	Duration (Months)	Serum Zinc (μg/dL)		Improvement Rate (%)	Year & Reference
					Before	After		
Zinc gluconate	55.1	35	67.5	3	80.5 ± 13.1	94.0 ± 24.6	77.1	1991
Plabebo	59.2	30	―		76.8 ± 10.8	71.9 ± 10.9	60.0	[39]
Zinc picolinate	55.2	37	87	3	71.0	81.6	75.6	2002
Plabebo	50.4	36	―		71.5	72.3	44.4	[40]
Zinc gluconate	61.1	24	20	3	72.28 ± 18.38	81.53 ± 19.61	50.0	2005
Plabebo	61.0	26	―		67.90 ± 14.64	72.01 ± 10.22	25	[41]
Poraprezinc	47.1	27	17	3	69.7 ± 12.1	Δ 5.7 ± 13.5	63.0	
	43.7	25	34		72.6 ± 13.9	Δ 11.4 ± 16.6	51.9	2009
	44.7	28	68		70.2 ± 11.7	Δ 20.6 ± 21.3	80.0	[42]
Plabebo	44.9	27	―		71.7 ± 13.6	Δ 1.8 ± 12.7	89.3	

Δ: rising value.

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
