# Peer review of "Japan’s Practical Guidelines for Zinc Deficiency with a Particular Focus on Taste Disorders, Inflammatory Bowel Disease, and Liver Cirrhosis"

_ijms, 2020, doi:10.3390/ijms21082941_

Round 1

Reviewer 1 Report

The reviewer has read the manuscript with interest. Authors summarized current situation regarding zinc deficiency in Japan, followed by involvement of zinc deficiency in dysgeusia, IBD, and chronic liver diseases. The reviewer has some comments.

  1. Title: The major content of this review article is about dysgeusia, IBD, and chronic liver diseases. Current title may lead to misreading by readers. Please reconsider about it.
  2. Fig. 1 and 2: These data are about Japanese populations. Please stipulate that in the figure legends. Additionally, dose marginal decrease of zinc uptake in both pregnant and lactating women have a significance?
  3. Table 1: All cited article seems to be derived from Japanese groups. Is such kind of analysis conducted only in Japan?
  4. Table 2: Please compare the Japanese guideline with other guideline issued by other country groups if available.
  5. Fig. 5 to 9: Are these figures required for this review article? Please explain clearly the relationship between the content from Fig. 5 to Fig. 9 and zinc (deficiency). If not, please delete these sections.

Author Response

Thank you for your response and the reviewers’ comments concerning our manuscript. We have revised our manuscript accordingly.

In the revised text, revised sentences showed in blue

Reviewer 1

Comment 1. Title should be reconsidered.

Response: Our title was revised to “Japan’s Practical Guidelines for Zinc Deficiency with a Particular Focus on Taste Disorders, Inflammatory Bowel Disease, and Liver Cirrhosis”.

Comment 2. The data of Fig 1 & Fig 2 should be stipulated the data of Japan.

  Does the data of Fig 2 (pregnant and lactating women) have a significant?

Response: First, we added “in Japan” to the figure titles for Figures 1 and 2, and to line 4 of the figure legend.

Secondly, as described in the figure legends, the recommended dietary allowance (RDA) for zinc (black bars) references the Dietary Reference Intake for Japanese, 2015 [4]. RDA is calculated as the amount considered sufficient for the majority of people, according to data from many studies. Data for zinc intake (white bars) references the National Health and Nutrition Survey 2016 in Japan [5]. Accordingly, statistical significance cannot be determined for differences between RDA and zinc intake. That said, standard deviation is shown for the intake data and appears as a fine bar. The top of the bar represents +SD; in other words, zinc intake is below the RDA in most pregnant and lactating women.

Comment 3. Table 1: Is such kind of analysis conducted only in Japan?

Response: No. This type of analysis has been reported in some studies from various countries.

As noted by Reviewer 2, biostatistical analyses were not performed, and Table 1 only references one or two previous studies. As such, we decided to remove Table 1 from the revised manuscript, and instead, added a description of diseases that often accompany zinc deficiency in section 1.1 of the main text.

Comment 4. Table 2: Please compare with other guideline by other country.

Response: To our knowledge, only one study has reported on practice guidelines for zinc deficiency: the BMJ Best Practice of Zinc Deficiency. We have added this study to the references for the present study and have added the following text:  

“To our knowledge, only one other report has been published on practice guidelines for zinc deficiency; namely, the BMJ Best Practice of Zinc Deficiency [26], which reports in detail on zinc deficiency, including symptoms, for serum zinc levels <60 µg/dL or 70 µg/dL. In this report, they also note that, due to the relatively low sensitivity of serum zinc levels in cases of marginal deficiency, oral supplementation should be considered if symptoms are typical, even if test results are normal; this is consistent with our diagnostic guidelines.”

Comment 5: Fig.5 to 9: Are these figures required for this review article? Please explain the relationship between the content from Fig.5 to Fig.9 and zinc (deficiency). If not, please delete these sections.

Accordingly, we have deleted Figures 8 and 9.  

Reviewer 2 Report

Kodama and colleagues offer a review of zinc deficiency regarding the Japanese population by providing discussions of (1) Japanese government-based public health nutritional analysis, (2) zinc as related to bitter taste function, (3) zinc and IBD, specifically with respect to macrophages, and (4) zinc with respect to liver function. The writing is well-done and easy to understand for each major section.  However, sections are not obviously linked, or are loosely relevant as currently presented. Specific issues that should be addressed include:

  • In the introduction, line 33, the statement "...and thus are being treated with medication." requires context.  What is meant here? Are there specific medications (chelators I assume) that they are referring to that are also specific to the thesis of the paper?
  • The authors state that the data in Table 1 "confirm" links to these disease states and zinc deficiency.  This is an overstatement in this manuscript as no specific hypothesis-driven work relevant to that table is  presented, nor attendant biostatistical analysis.
  • Lines 91 and 92.  This reference needs to be more specific so the reader can more easily find that information.
  • Lines 110-111. Medication of disease states are mentioned, I encourage the authors to discuss specific medications and their tested relationships to zinc effect.
  • Section 3 seems to be like a standalone mini-paper. It is not clearly linked to sections 1 & 2.  Additionally, sections 3.2 and 3.3 (and Fig. 3) appear to contain a lot of original observations and discussion of data that are not yet published/referenced. This is an awkward placement for a review article, especially one that has extensive discussion of literature at the public health and patient levels.
  • Most of section 4 appears unnecessary as it is mostly a broad review of liver amino acid metabolism rather than specifics about zinc. While the introduction to section 4 vaguely links zinc to liver metabolism, 4.1, 4.2, 4.3 and relevant figures are superfluous to this paper, containing broad restatements of another review published by them (and indeed these sections do not even discuss zinc).  I strongly encourage expanding the groundwork set by the brief review of literature in sections 4.4 and 4.5 should the authors decide to keep this section.
  • The references in the last paragraph of section 4.5 are missing.
  • Figure 12 is not necessary as this is already published.
  • Second paragraph of Conclusions is not necessary as this was stated several times already including in a table.

Overall, the writing of individual sections is very good here, but I encourage the authors to focus the paper on reviewing relevant zinc biology throughout, and by very clearly linking each section. This topic is very interesting, especially as it relates to public health level information, clinical trials, and mechanisms underlying inflammation.  There is a lot of literature on the section topics presented by the authors, especially zinc and macrophages and zinc and taste, that is not addressed by the present draft.

Author Response

Reviewer 2

Comment: Introduction:line 33, “—and thus are being treated with medication” What is meant?

Response: We have removed this sentence to avoid any confusion.

Comment: Table1 This is an overstatement.

Answer: As you note, biostatistical analyses were not performed, and Table 1 only references one or two previous studies. As such, we decided to remove Table 1 from the revised manuscript, and instead, added a description of diseases that often accompany zinc deficiency in section 1.1 of the main text.

Comment Lines 91 and 92:

Response: We have added a website for the Food Data Chart for zinc.

Comment line 110-111. Medication of disease state:

Response: This sentence has been removed.  

Comment: Section 3 seems to be like a standalone mini-paper. It is not clearly linked to sections 1 & 2.  Additionally, sections 3.2 and 3.3 (and Fig. 3) appear to contain a lot of original observations and discussion of data that are not yet published/referenced. This is an awkward placement for a review article, especially one that has extensive discussion of literature at the public health and patient levels.

Response: Our present review does not include any original or unpublished data. All previous studies are carefully referenced, including those from our group. In this section, we have included the results from the basic research studies, but have included these in a form that emphasizes the significance of these data in clinical practice.

Comment: Most of section 4 appears unnecessary as it is mostly a broad review of liver amino acid metabolism rather than specifics about zinc. While the introduction to section 4 vaguely links zinc to liver metabolism, 4.1, 4.2, 4.3 and relevant figures are superfluous to this paper, containing broad restatements of another review published by them (and indeed these sections do not even discuss zinc). I strongly encourage expanding the groundwork set by the brief review of literature in sections 4.4 and 4.5 should the authors decide to keep this section.

Response: Accordingly, we have removed sections 4.1, 4.2, and 4.3, as well as relevant figures (Figures 8 through 12).

Comment: The references in the last paragraph of section 4.5 are missing.

Response: Accordingly, two references have been added in the last paragraph of section 4.5.

Comment: Figure 12 is not necessary as it is already published.

Response: We have removed Figure 12.

Comment: Second paragraph of conclusion is not necessary.

Response: We have shortened the second paragraph, as follows:  

“The Japan Practice Guideline for zinc deficiency, was issued. Although the serum zinc level is a criterion, serum zinc levels does not appear to accurately reflect zinc deficiency. Accordingly, more sensitive and accurate biomarkers of zinc deficiency are needed,”

Round 2

Reviewer 1 Report

Manuscript was much improved through a revision. The reviewer has no further comment.

Reviewer 2 Report

The authors have faithfully improved the manuscript per my original review. I have not further criticisms for the authors.